## Original Research Article

calibration; controlled environment plant growth; dynamic light; LED; light; light recipe; realistic conditions.

**Corresponding author:**
Daphne Ezer;
Email: daphne.ezer@york.ac.uk

**Associate Editor:**
Prof. Boon Leong Lim

# Accurately programming complex light regimes with multichannel LEDs

Gina Y. W. Vong, Paul Scott, Will Claydon, Jason Daff, Katherine Denby and Daphne Ezer [ID]

Department of Biology, University of York, UK

## Abstract

Advances in LED lighting technologies enable increasingly complex light regimes, providing greater insight into plants' responses to dynamic light – such as seasonality and fluctuating conditions – rather than the traditional discrete (i.e., on/off) lighting. However, current methods of programming such regimes are time-consuming and/or limited to 1–2 wavebands. Robust methods are therefore needed to accurately programme multichannel/waveband LED lighting systems. We present a multistep, multidimensional algorithm to accurately programme multi-waveband LED lights. This algorithm accounts for non-linearity between intensity settings and measured light quantity output, as well as optical crosstalk between channels of different wavebands. It outperforms methods that treat waveband channels as independent variables, allowing users to more accurately programme multichannel light regimes. This will allow the community to probe plant responses to dynamically changing light spectra. We have made this algorithm available as an R package, LightFitR (installable from CRAN with 'install.packages("LightFitR")'.

## 1. Introduction

Light is an essential source of information for plants (Su et al., 2017) to optimise their photosynthesis. Red to far-red ratios drive shade avoidance (Ballaré & Pierik, 2017; Franklin, 2008; Ruberti et al., 2012), while blue light drives photomorphogenesis (Hajdu et al., 2018; He et al., 2019; Sun et al., 2013; Xu et al., 2021). Plants have evolved elaborate mechanisms for gathering this information, integrating it with other environmental inputs and translating it into a physiological response (Alvarez-Fernandez et al., 2021; Annunziata et al., 2017; Balcerowicz et al., 2021; Song et al., 2018). Much work has been done to study the mechanisms of how plants perceive and respond to light information. These insights allow us to understand, with increasing resolution, how a plant behaves, with the potential to inform strategies to engineer climate-resilient crops and novel agricultural technologies such as indoor vertical farms (Bechtold et al., 2025; Paradiso & Proietti, 2022).

Light information can be subdivided into three components: (i) light quantity, how much light the plant is receiving (Poorter et al., 2019); (ii) photoperiod (Goto et al., 1991), the duration for which the plant is receiving light; and (iii) spectral quality, corresponding to the spectrum and wavelengths or colours of light (Paik & Huq, 2019). Together, these three components allow the plant to infer information about the time of day (Paajanen et al., 2021), seasons (Gangappa & Kumar, 2018; Nieto et al., 2022; Ronald et al., 2024) and shade (Ballaré & Pierik, 2017).

However, much of the work on plant light response uses discrete lighting changes (e.g., sudden on-off switches) or constant light/darkness (Alvarez-Fernandez et al., 2021; Balcerowicz et al., 2021), rather than the dynamic changes seen in nature where light quantity, photoperiod and spectrum often gradually change over the day and across seasons (Poorter et al., 2016). Increasingly, research has focused on plant responses to fluctuating light conditions throughout the day (Annunziata et al., 2017; Belmonte et al., 2024; Emmerson et al., 2024; Kaiser et al., 2018; Mehta et al., 2024; Song et al., 2018; Vialet-Chabrand et al., 2017), giving us a deeper understanding of plants' perception and response to light signals. These studies take advantage of advances in LED technologies (Stevens et al., 2024; Yano & Fujiwara, 2012), which allow

researchers to modulate the intensity across multiple waveband channels in pre-defined schedules to produce dynamic light regimes. However, these existing studies focus on fluctuating light quantity, using broad-spectrum LEDs or 1–2 waveband channels only. Despite strong evidence that plants perceive ratios/combinations of wavelengths (e.g., R:FR for shade avoidance (Ruberti et al., 2012)), which are known to change throughout the day (Holmes & Smith, 1977; Kotilainen et al., 2020), few studies explore plant responses to dynamic spectrum quality due to a lack of methods for accurately programming complex light regimes involving multiple waveband channels,. Further, there is currently inconsistent reporting of lighting regimes used in these experiments (Both et al., 2015).

Existing methods for calibrating and programming LED lights vary. Some use regression methods (Hashida et al., 2022) while others use laborious 'measure and adjust' approaches (Thomas et al., 2020). Regression-based methods assume a linear relationship between intensity (settings within the LED, typically a percent) and the measurable light output from the LED. By treating each LED channel individually (i.e., assuming independence between the channels), neither approach accounts for the complex interaction of the spectra produced by multiple overlapping channels. A spectrometer, rather than a light meter, allows the assessment of the wavelength range of each LED channel, the extent of overlap between channels and whether a channel has a secondary peak (Both et al., 2015; Holmes, 1984b). All these factors can influence the output of the LED fixtures and, subsequently, plant responses in experiments (Both et al., 2015).

Moreover, the output of the lights is highly dependent on their installation, as different materials on the growth cabinet walls have differing reflectance and absorbance. This affects how much and which wavelengths of light reach the plant. Therefore, each LED lighting setup requires a bespoke calibration to accurately programme a light regime for experiments.

Here, we present an experimental protocol coupled with an algorithm for effectively calibrating and programming complex light regimes involving multiple LED waveband channels across a range of intensities. We also present a further refinement step for those requiring additional accuracy. The algorithm is available as an R package, LightFitR, facilitating the employment of more complex light regimes for plant science experiments.

## 2. Methods

All data processing, analysis and figure creation were carried out in R (version 4.4.1) (R Core Team, 2024). The code is available on GitHub at: https://github.com/ginavong/2024_LightFitR_MethodsPaper, and the raw data are available at: https://doi.org/10.5281/zenodo.15584172

### 2.1. Defining key terms

In the context of this manuscript, we use the term *LED channel* to denote the waveband produced by a given type of LED. Meanwhile, *LED fixture(s)* refer to the lighting fixture as a whole, including all the LED channels housed within.

We use the term *intensity* to refer to the unitless, discrete/integer setting we provide to each LED channel. This is the term often used by LED light manufacturers to denote the settings, and functions similarly to the 'volume' setting in audio equipment. In contrast, we use the term *photon flux density (PFD)* to refer to the instantaneous measurement (area by time) of light output that is present at specific designated wavelengths (in nm) at the plant level in the growth chamber. The aim of our protocol is to enable a researcher to predict what intensity they should programme for each LED channel to produce a *target PFDe* at a target waveband required for a specific experiment.

*Peak wavelength* is the precise wavelength where the maximum PFD is observed for that channel. We define *optical crosstalk* as the PFD of light from one LED channel, which is encroaching on the peak wavelengths of other channels.

Further, we use the term '*event*' to refer to the period when a given set of LED channels is on at a given set of intensities. A series of events is referred to as a *regime* or *schedule*.

### 2.2. LED fixtures

We used Heliospectra DYNA (Gothenburg, Sweden) LED fixtures, which have nine individually programmable LED channels, which are dispersed throughout the lamp's LED array: 380, 400, 420, 450, 530, 620, 660 and 735 nm, and a broad-spectrum 5,700 K 'cool white' channel (Figure 1a). The LED channels were controlled by helioCORE™ (R3.2.2-Release), a web-based lighting control system, independent of the growth chamber's local control system. Each waveband channel can be set to its own integer 'intensity' between 0 and 1,000, with the ability to set schedules such that each channel can be set to different intensities at a given time of day.

This lamp model uses pulse width modulation (PWM) dimming at intensities below 200, which creates on/off pulses. Constant current dimming is used above intensities of 200.

### 2.3. Cabinet setup

The experiments were conducted in a Snijders (Tilburg, The Netherlands) MC1750 growth chamber with internal dimensions (w × d × h) 1,830 mm × 780 mm × 1,230 mm. The chamber was modified to provide two equal growing compartments with separate lighting regimes by installing an internal partition made from white Palfoam (Palram, Hull, UK) aerated PVC foam sheet. The partition was sealed with RS PRO (Corby, UK) 5 mm Black Foam Tape to prevent light spillage between the compartments and did not impair the upward airflow distribution system (Figure 1b). Sufficient outdoor make-up air provides ambient $CO_2$ conditions inside the chamber. The chamber air temperature was maintained at 22/22°C (s.d. ± 1/1°C) during the light/dark period. The relative humidity in the chamber was maintained at 55% (s.d. ± 10%).

Heliospectra DYNA (Gothenburg, Sweden) LED fixtures, with external dimensions 425 mm × 199 mm × 219 mm (w × d × h), were mounted in the existing passively ventilated lamp loft. These LED fixtures were partitioned by a clear glass barrier above the growing compartments. The LED fixtures were cooled by internal variable-speed fans.

### 2.4. Light measurements

For automated light measurements, schedules were uploaded to the LED fixtures via helioCORE™. Each event in the schedule lasted 5 min to allow time for multiple readings to be taken.

Spectral data were collected at the plant growth level, 95 cm below the centre of the LED fixture (Figure 1c). For spectrum readings, an OceanInsight FLAME spectrometer (Ocean Optics, Duiven, The Netherlands) was used in conjunction with Ocean-View (version 2.0.14) software, recording every 1 min with a 500 ms integration time and averaging over 5 scans.

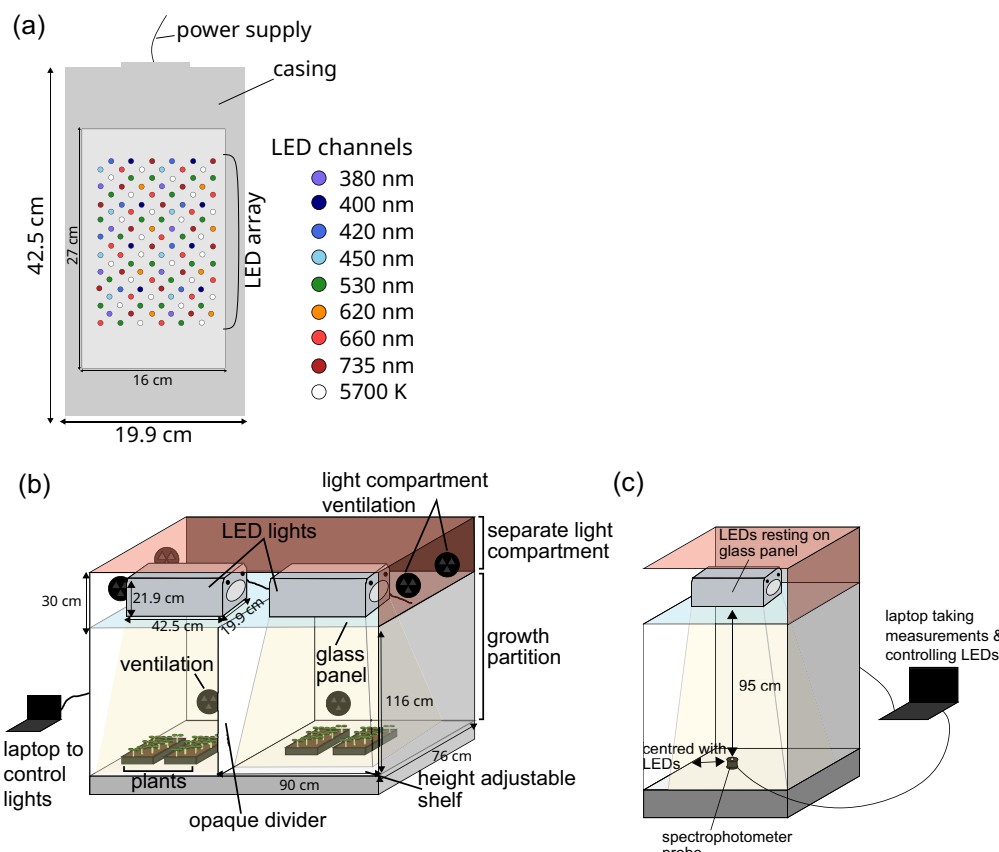

**Figure 1.** Cabinet setup with Heliospectra DYNA lights. (a) Layout of LED channels within the lighting fixture. (b) Overall setup of the growth cabinet with two Heliospectra DYNA lights resting on a glass shelf in a separate lighting compartment with its own ventilation. These lights are connected to a laptop, which controls them. The growth compartment is separately ventilated, with a height-adjustable shelf for plants. An opaque divider was installed to separate the light coming from the two sets of LEDs above. (c) Setup for taking measurements. Only half of the cabinet is shown. The spectrometer rests on the shelf, 95 cm away from the light source. The spectrometer is connected to a laptop, which records the measurements while also controlling the LED lights.

## 2.5. Spectrometry data processing

The spectrum measurements were processed to trim the wavelengths from 300 to 800 nm. Unit conversions from $\mu W\ cm^{-2}\ s^{-1}\ nm^{-1}$ to $\mu mol\ m^{-2}\ s^{-1}\ nm^{-1}$ were calculated using the *photobiology* package (version 0.11.3) (Aphalo, 2015) in R. As there were multiple readings for each event, the reading at the middle time point was selected for subsequent analysis. Further annotation of the processed dataset was then carried out depending on the analysis required (see specifics below).

## 2.6. Generating and analysing calibration data

Calibration curves were generated separately for each LED channel (380, 400, 420, 450, 530, 620, 660 and 735 nm and 5,700 $K$ white), by increasing the intensity every 5 min (0, 1, 5, 10, 20, 50, 100, 200, 300, 400, 500, 600, 700, 800, 900 and 1,000) and taking spectrometer readings every 1 min (see Section 2.4.).

After the raw measurements were processed (see Section 2.5), the following values were calculated for each reading (Figure 2a): (i) the total PFD (i.e., sum of PFD between 300 and 800 nm); (ii) the peak wavelength (i.e., wavelength at which the maximum PFD occurred in the spectrum), with the median peak wavelength for the LED channel used for future analysis; and (iii) the PFD at the peak wavelength (i.e., peak PFD). Further, we calculated *optical crosstalk* for each channel at 1,000 intensity by extracting the irradiances at the peak wavelengths of the channels that were not active.

## 2.7. Intensity prediction algorithm

Our algorithm (Figure 2b) takes user-provided calibration data and user-defined target irradiances for each peak wavelength to predict intensities for each channel. These intensities can be programmed into the fixtures to achieve the target regime. This algorithm is suitable for light fixtures with at least two different waveband channels, and is unit-agnostic provided the units are consistent. However, it has not been designed for broad-spectrum LEDs, such as the 5,700 K channel. Our algorithm consists of three steps: (i) produce a rough estimate of intensities by directly using the calibration data for each LED channel, (ii) adjust the estimated intensities to account for optical crosstalk between LED channels, and (iii) modify estimated intensities to fit hardware requirements (Figure 2b).

First, when given an event of target PFDs at the peak wavelengths, it searches the calibration data (example in Supplementary Table S1) for intensities that produce the closest PFDs to the target irradiance for each channel (i.e., the intensity with the smallest absolute difference in PFD, called the *closest intensities*. We define $m$ to be the vector of closest intensities for each LED channel.

Next, we refine these intensities to account for optical crosstalk between channels (see Section 2.6). We combine optical crosstalk at the closest intensities identified in the first step into a matrix, $M_m$, where rows represent peak wavelengths and columns represent LED channels, such that each cell is the PFD at a particular wavelength when the channel is at the closest intensity. This matrix

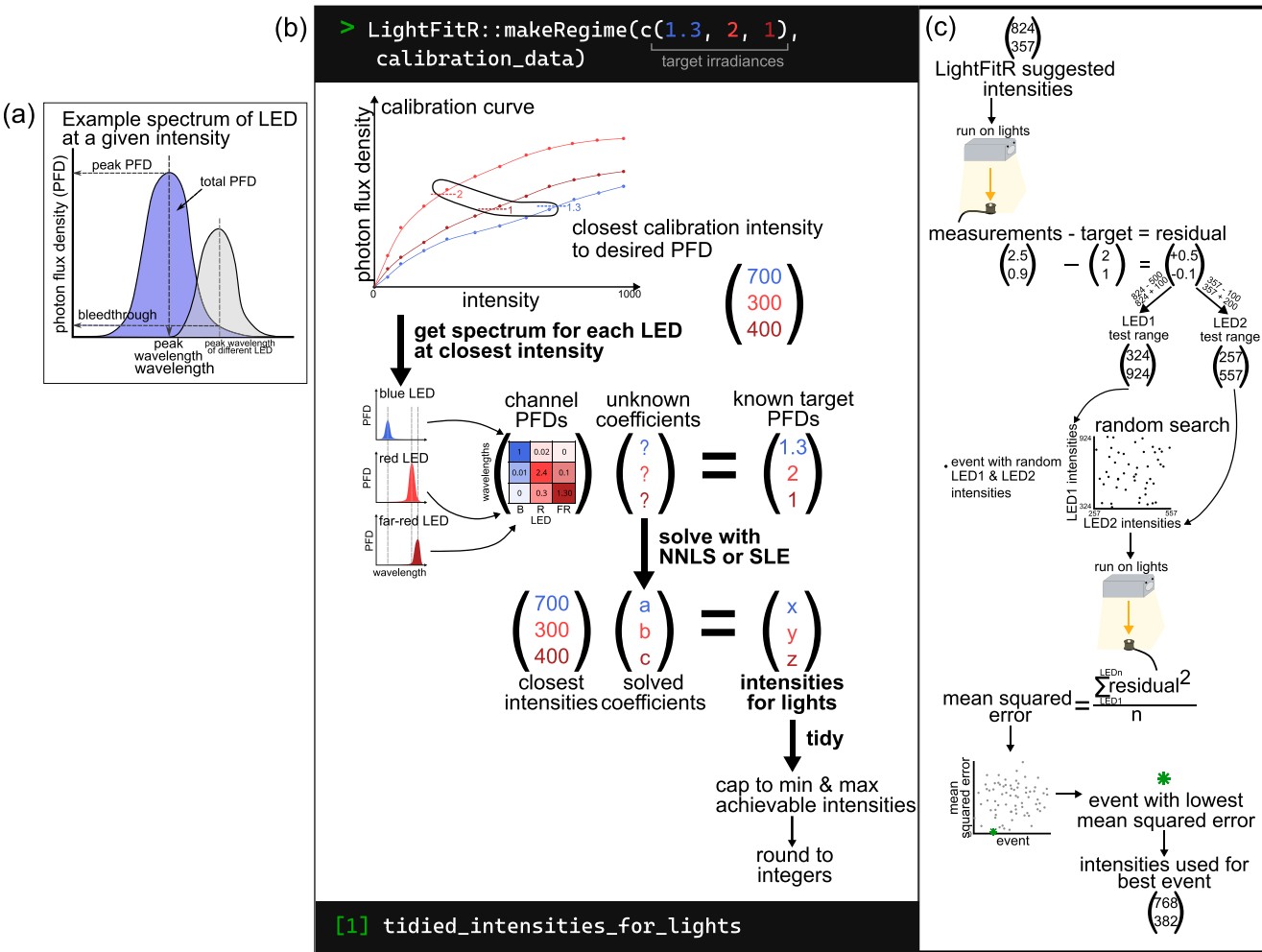

**Figure 2.** Analysis and algorithm diagrams. (a) Example spectrum from an LED at a given intensity, breaking down which features were used for analysis: peak wavelength, peak PFD, total PFD, bleedthrough. (b) Graphical explanation of our algorithm for accurately programming LED lights. It takes user-provided targets and calibration data and uses them to predict which intensities to set the lights in order to achieve the targets. This is achieved through multiple steps: obtaining the closest calibration intensity to the target, getting the calibration spectrum for each LED at that closest intensity, compiling a matrix of these channel PFDs, solving a system of linear equations with this matrix and the targets, multiplying the solved coefficients with the closest intensities to predict the optimum intensities for the lights and tidying the predicted intensities so that they are accepted by the lighting software. (c) The process of carrying out a random search. Take the intensities that the LightFitR algorithm predicts and run it on the lights, measuring their output. Calculate residuals by comparing the measurements with the original targets. Test ranges for each LED are calculated proportionally to the scale and direction (positive or negative) of the residual, using the LightFitR intensities. Random combinations of intensities (within these ranges) are selected and then run on the lights, with each random combination considered an event. Measurements are again taken, and the mean squared error for each event is calculated using the residuals. The event with the lowest mean squared error is the 'best' event, and the intensities used for that best event should be used in experiments.

is then used to solve a multidimensional system of linear equations, either using R's 'solve' function (we simply call this approach 'system of linear equations (SLE)') or with non-negative least squares (NNLS, Lawson & Hanson, 1995), as in Equation 1. To calculate the predicted intensities, we use Equation (2).

$$M_m x = b \qquad \text{(Equation 1)}$$

where $M$ is the channel irradiance matrix at LED intensities $m$, $x$ is the vector of coefficients we are trying to predict and b is the vector of target irradiances.

$$mx = c \qquad \text{(Equation 2)}$$

where $c$ is the predicted intensities that we expect will produce irradiances close to the target irradiance $b$.

Finally, we must adapt these estimates to meet the hardware requirements of the LEDs. For example, these predicted intensities may be below 0 (in the case of SLE) or above the maximum achiev-

able intensity by the LEDs (1,000 for Heliospectra DYNA). Furthermore, the LED fixtures can only take integer (whole numbers) intensity values, while the predicted intensities are often not integers. Therefore, we created a final 'tidying' step, which (i) rounds to the nearest integer; (ii) sets negative predicted intensities to 0; and (iii) caps the maximum intensity to the maximum achievable by the light model. Example output is in Supplementary Table S2.

## 2.8. Algorithm comparison

We compared our multidimensional algorithm (with SLE and NNLS) with other methods: individual closest, individual linear model/regression and individual NNLS.

We designed a test regime where random LED channels were turned on in combination, at random intensities. Our testing regime increased in complexity, starting with two random channels active/on at a given time and increasing until all eight LED channels

were on simultaneously, with *n* = 20 events for each level of complexity (Supplementary Table S2). We ran this regime and measured the spectra for each event (see Sections 2.4 and 2.5).

Using the measured PFD, we asked each algorithm to predict the intensities that were originally used. This allowed us to assess the accuracy of the algorithm predictions for each LED channel at each event (i.e., the residuals = predicted − true). To produce an overall score for how accurate each algorithm was for a given event, the mean squared error (MSE) across LED channels was calculated for each event. We compared the algorithms at each complexity level using a Kruskal–Wallis test with Dunn post-hoc test and Holm's *p*-value adjustment. To assess whether an algorithm was consistently over- or under-shooting, we created visualisations of the raw residuals.

### 2.9. Refinement with random search

Additionally, we suggest experiments for further refining the intensities to produce PFDs that are even closer to the target by using a random search approach (Figure 2c). The premise is that we select random combinations of intensities near our algorithm's predicted intensities, which can be experimentally tested on the LEDs to enable further refinements. The range of the intensities is weighted by how 'wrong' the algorithm-predicted intensities are. Example code for creating a random search is available on GitHub: https://github.com/ginavong/2024_LightFitR_MethodsPaper/blob/master/R/regime_design/5.2_RandomSearch_regime.R

More specifically, we programme the algorithm's predicted intensities in our LEDs (i.e., the starting intensity), measure the PFD at each waveband and calculate the difference between the actual PFD and the target PFD (i.e., the residual). We defined test ranges for each LED channel that were proportional to the direction and size of the residual for the algorithm-predicted intensity (see Equations 3 and 4). We then created events using random combinations of intensities within each channel's test range. These events were tested on the LEDs and the spectra collected. The optimal intensities are then defined as the combination of intensities that have the lowest MSE.

$$bound1 = i - (p_1 \epsilon s) \qquad \text{(Equation 3)}$$

$$bound2 = i + (p_2 \epsilon s) \qquad \text{(Equation 4)}$$

where *bound1* defines an intensity limit in the opposite direction of the residual ($\epsilon$) and *bound2* defines an intensity limit in the same direction as the residual. *i* is the algorithm predicted intensity, and *s* is a scaling factor that turns the small residuals (in $\mu$mol s$^{-1}$ m$^{-2}$) into numbers >1 (we used *s* = 100 but other setups may differ). *p* is a proportionality factor that indicates how far away from the algorithm-predicted intensity should be searched, where $p_1 > p_2$ to ensure that the range of *bound1* is greater than *bound2* (we used $p_1 = 10^{1.4}$ and $p_2 = 10$).

### 2.10. Testing the efficacy of random search

We tested the random search approach for three different sets of target PFDs to use as 'treatments'. This allowed us to assess the marginal benefit of using random search for different starting (i.e., algorithm-predicted) MSEs. To determine the treatments, we used targets from the Section 2.8 at a complexity of four LED channels, selecting the events with the lowest (minimum), median and highest (maximum) starting/algorithm-predicted MSE. *n* = 50

events were selected for each treatment, each with a random combination of intensities within the LED channels' test ranges.

After processing the raw spectrometer data, the dataset was filtered to retain only the PFD at the median peak wavelengths for further analysis. These were compared with the target PFD to calculate residuals and then MSE per random search event. Within each treatment, the random search event with the lowest MSE was labelled the 'best' event. The Euclidean distance between each random search event and the corresponding 'best' event was calculated, and Spearman's rank was used to assess the correlation between MSE and Euclidean distance.

### 2.11. Simulating R:FR from natural conditions

To assess LightFitR's ability to approximate more 'natural' conditions, we attempted to recreate the R:FR shown in Kotilainen et al. (2020) where the authors measured the R:FR on a cloudy afternoon in Helsinki, Finland, on 11 August 2017. To achieve this, we took approximate R:FR from the figure at every 5° of solar elevation from 10 to 40, and every 2.5° between −2.5 and 10. We then calculated the peak red and far-red PFDs to achieve these R:FR, and used those as the targets for the LightFitR algorithm. Our algorithm predicted intensities, we ran them on the lights, and measured the spectra before calculating the measured R:FR. This was plotted on a graph and compared with the target R:FR from the figure.

## 3. Results

### 3.1. Properties of the lighting system

'Intensity' is an arbitrary unit within the light system settings, associated with the amount of electrical power that a particular channel receives. We investigated how intensity impacts the amount of light (photon flux density (PFD), $\mu$mol m$^{-2}$ nm$^{-1}$ s$^{-1}$) being output. We noted that irradiance does not scale linearly with intensity (Figure 3a and Supplementary Figure S1a). Each LED channel has a different maximum PFD, with the 380 nm channel emitting particularly low light. Moreover, the maximum PFD for the 620 nm channel was around 600 intensity rather than 1,000.

Also, we noticed that the spectra of some LED channels overlapped with each other (Figure 3b and Supplementary Figure S1b), resulting in optical crosstalk. The broad-spectrum 5,700 K channel has two peaks, one of which overlaps directly with the 450 nm channel (Supplementary Figure S1b), so it was excluded from further analysis and algorithm testing. This optical crosstalk is particularly severe among the blue (400, 420 and 450 nm) channels. Optical crosstalk was also evident among the red (620, 660 and 730 nm) channels. Also, there is a small amount of optical crosstalk between the blue and red channels and vice versa. The algorithm we developed accounts for both the non-linear associations between intensity and PFD, as well as the optical crosstalk between LED channels.

Finally, we wished to confirm whether the peak wavelength for each channel matched the named wavelength of the channel (Figure 3c). None of the channel peaks precisely matched the purported wavelengths, but some channels deviated more than others. At low intensities, the noise from the spectrometer was greater than the PFD at the peak wavelength. Thus, the peaks around 800 nm at those low intensities do not represent true output from the light. However, at intensities over 100, the peak wavelengths were changing and diverging from the purported wavelengths as

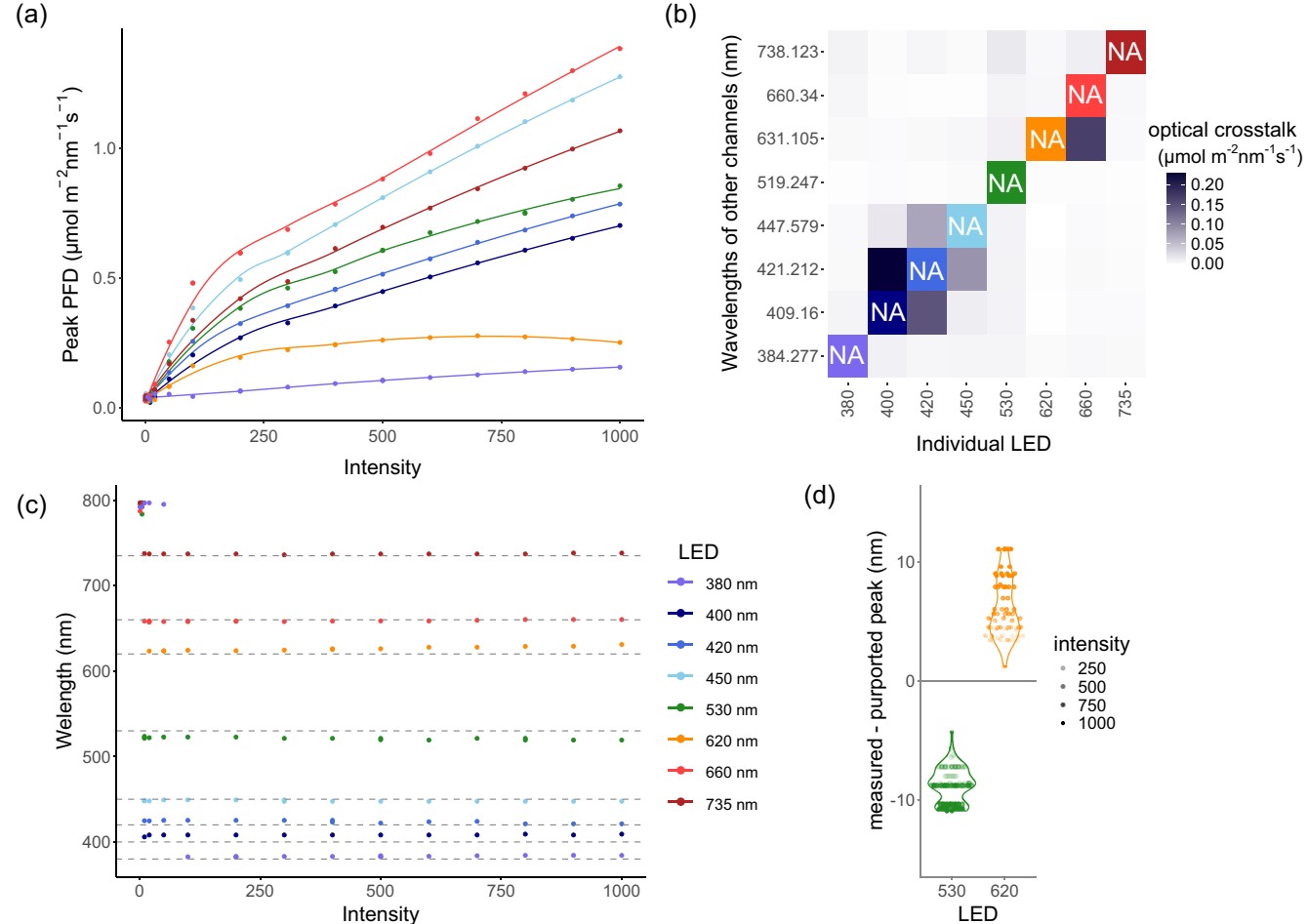

**Figure 3.** Challenges of the lighting system, revealed during calibration. The spectrum was measured independently for each LED channel at increasing intensities (0, 1, 5, 10, 20, 50, 100, 200, 300, 400, 500, 600, 700, 800, 900 and 1,000) with a spectrometer. (a) The irradiances at the peak wavelengths of the spectrum, with a smoothing function applied to highlight the non-linearity. (b) The bleedthrough from one LED into the wavelengths of the other channels, at maximum (1,000) intensity. The diagonal is NA, as that represents the channel that is on. (c) The wavelength of the spectrum peak for each LED channel, with dashed lines representing the purported peak wavelengths. (d) The difference between the peak we measured and the purported peak, and how it relates to intensity. The 530 nm and 620 nm channels are highlighted as they show the most extreme differences.

the intensity increased (Figure 3d). Going forward, we define the peak wavelength for a channel as the median of that channel's peak wavelengths across all intensities.

### 3.2. Performance of the prediction algorithm

Our algorithm aims to accurately predict intensities for LED lights to achieve user-defined target PFDs and wavelength profiles. To test the performance of our algorithm, we collected spectral data from a complex light regime that had random combinations of LED channels. Using these measurements (PFD and wavelength) as targets, we asked the algorithm to predict the intensities needed for each LED channel to achieve the measurements. These predicted intensities were then compared to the original, true intensities (i.e., predicted – true intensity) to assess the accuracy of the prediction for each LED channel. Mean squared error (MSE) was used to summarise the accuracy of the predictions across all LED channels for a given event. To benchmark our algorithm, we compared our multidimensional approach – either with NNLS or SLE – with three other approaches that treat each LED channel individually: (i) taking the closest intensity from the calibration data (Thomas et al., 2020); (ii) carrying out a separate linear regression

across each LED channel (Hashida et al., 2022); and (iii) computing a non-negative least squares regression across each channel individually.

We observe that our multidimensional approaches consistently show lower MSEs compared to the approaches that treat LED channels individually (Figure 4a) ($p < 0.01$, Supplementary Table S3), with a particular advantage as the complexity (i.e., number of active LED channels) increases. We note that multidimensional SLE and multidimensional NNLS produce very similar MSEs for this test.

We investigated whether the final tidying step of our algorithm impacted the accuracy. The pre-tidied (i.e., raw predicted values) predictions did not have a substantial difference in MSE (Supplementary Figure S2) compared with tidied predictions.

When comparing linear regression (the best of the individual approaches) and multidimensional NNLS (the best multidimensional approach) (Figure 4b), we see that both consistently predict higher intensities than the true intensity in the blue (380, 400, 420 and 450 nm) channels, but this effect is less pronounced with the multidimensional approach. We hypothesise that this is due to our approach being able to account for the interaction between LED channels, particularly in the blue range (B), where there is significant optical crosstalk from overlapping spectra.

(a)

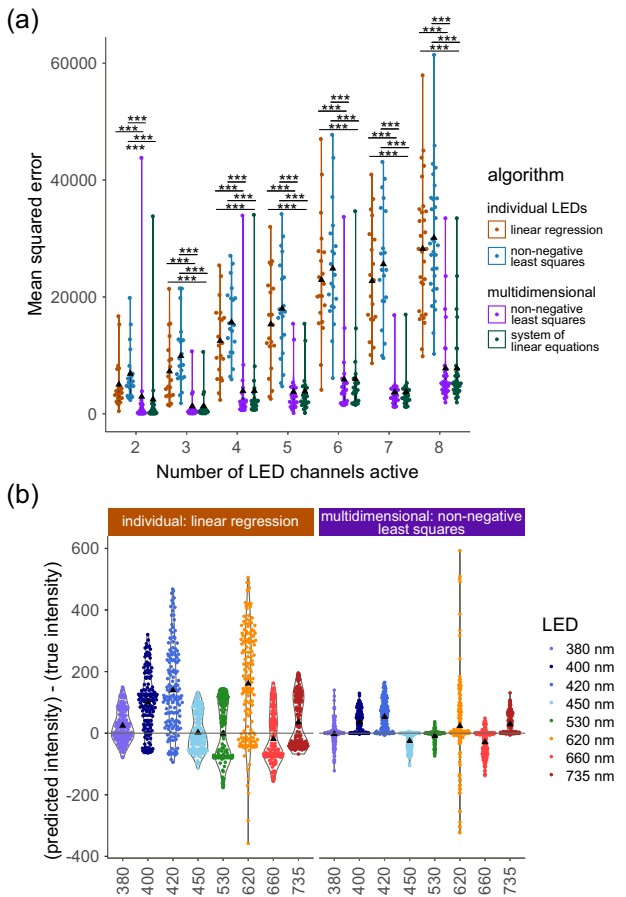

(b)

**Figure 4.** Testing our algorithm against other methods. The algorithms were tested on their ability to predict the correct intensities, which were originally used in a light regime with random intensities in random combinations of LEDs at increasing complexity. Individual LED approaches (closest, linear regression and non-negative least squares) were compared with multidimensional methods (system of linear equations, non-negative least squares). ▲ denotes the mean. (a) Mean squared error of each event at increasing complexities. For all algorithms, see Supplementary Figure S2. (***) indicates $p < 0.001$ in pairwise comparisons within the complexity level. (b) Residuals ([predicted intensity] − [true intensity], i.e., how wrong the prediction is) per LED across all complexity levels. The best individual approach is compared with a multidimensional approach to illustrate the differences. For the LED breakdowns across all algorithms, see Supplementary Figure S3.

We note that the near-red (620 nm) channel performs consistently poorly across all algorithms. We hypothesise that this is due to the non-monotonic relation between intensity and PFD (Figure 3a).

### 3.3. Refinement using random search

To assess whether our algorithm's predictions could be improved, we carried out an additional refinement process using a random search. Here, intensities within a defined range are randomly selected to search for a PFD closer to the target (Supplementary Figure S4a). We selected targets where our algorithm-predicted intensities generated low, medium and high starting MSEs to assess the marginal benefit of this additional refinement.

Regardless of the starting MSE, the random search was able to improve the accuracy of the lights (Figure 5). The high starting MSE seems to have benefited most from this refinement process. Users

(a)

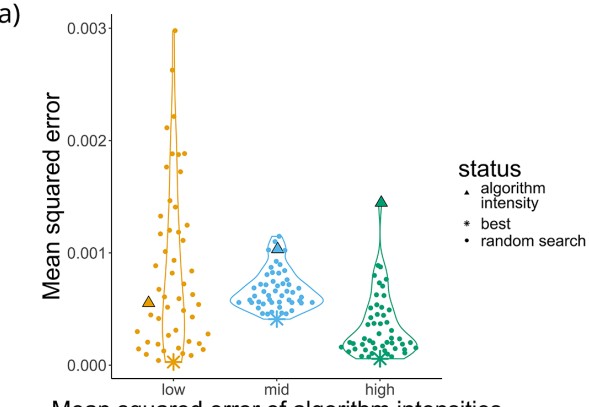

(b)

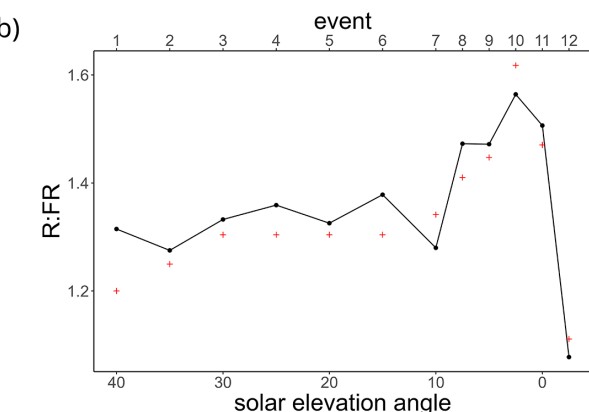

**Figure 5.** (a) Random search. Target irradiances (complexity: four LEDs) were chosen based on the accuracy of the intensities predicted by the multidimensional NNLS algorithm, with a low, medium and high mean squared error (of the algorithm prediction) selected (▲). These algorithm-predicted intensities were used as the basis of a random search in an attempt to find the light intensities with the lowest mean squared error (∗). Each point represents an event where random intensities (within a range) were selected for each of the four LED channels. (b) Simulation of the R:FR ratios observed across an afternoon in Helsinki on 11 August 2017 from Figure 3f of Kotilainen et al. (2020). Black lines represent measurements from the lights, programmed using our algorithm. Red + represents the outdoor measurements, that is, the targets.

may not need to invest extra time in refinement if the intensities predicted by the algorithm are already very close to the target.

The random search for the mid starting MSE treatment was not able to reduce the MSE as much as the low and high treatments. We hypothesise that the challenge in identifying intensities in this treatment comes from the inclusion of the 620 nm channel (Supplementary Figure S4a), which had a non-monotonic relation between intensity and PFD (Figure 3a).

When taking measurements using any instrument, there is an amount of random technical noise that affects the accuracy of the measurement. Since the MSEs we observe are quite small, we wished to determine whether the improved accuracy from the random search arose because we were truly approaching the local optimum intensity rather than capturing random noise in spectrometer readings. To assess this, we determined whether the random search intensity combinations that were more similar to our final 'best' intensities (i.e., small Euclidean distance) produced more accurate PFD (i.e., lower MSE) than random search intensities that were further away (i.e., large Euclidean distance). If our results are due to random technical noise, we would expect a random distribution of points in Supplementary Figure S4b. However, in the mid and

high starting MSE treatments, Supplementary Figure S4b shows a smooth gradient in MSE with a strong positive correlation (Supplementary Table S4), therefore confirming that we are approaching the local optimum for those treatments and that random search is an effective strategy for refining selected intensities. To investigate the possible fluctuations of the lights or spectrometer measurements, we produced a spectrum and time series where all eight LED channels were on simultaneously (Supplementary Figure S5).

## 4. Discussion

Most research on plant light response has been done under discrete conditions that are dissimilar to the dynamic natural light conditions they have evolved under (Holmes & Smith, 1977; Poorter et al., 2016). Though it is not possible to fully reproduce natural daylight indoors, advances in LED technology allow us to programme more realistic light regimes, which mimic parts of the spectrum and fluctuations of natural daylight at different latitudes and times of year (Figure 5b, Kotilainen et al., 2020). Recent research has highlighted the value of performing experiments using more complex light regimes. For instance, fluctuating light affects leaf morphology and photosynthetic acclimation (Vialet-Chabrand et al., 2017). Moreover, natural light results in differences in metabolite use (Annunziata et al., 2017). As a third example, twilight length affects growth and flowering (Mehta et al., 2024). Taken together, these studies show that conclusions from discrete lighting experiments cannot be directly translated to natural conditions. Therefore, dynamic lighting conditions are potentially a fruitful area for further research, giving a more nuanced understanding of plant response to light.

However, the complexity of the light regimes that are possible to investigate is limited by the difficulties of accurately programming LEDs across multiple channels. The algorithm we present here allows us to more accurately programme LED fixtures, unlocking the potential of these LEDs for research. By using a multistep approach, LightFitR (https://github.com/ginavong/LightFitR) can account for a non-linear relationship between intensity settings and measured light output and channels with overlapping wavelength ranges. Though our approach has not yet been tested on other LED-based models, the package has been designed to be light-agnostic. Our package outputs predicted intensities in a human- and machine-readable format (e.g., in Supplementary Table S2), meaning that it can be adapted for any programmable LED-based lamp models, which have multiple waveband channels. This package, therefore, allows the community to more accurately programme complex light regimes involving multiple LED waveband channels, fine-tuning wavelength ratios, as well as gradual changes to light quantity through the day.

We have shown the value of calibrating the lights when installing them in a cabinet (Holmes, 1984a) and suggest calibrating them regularly thereafter. Although some LEDs in some scenarios show more linearity between intensity and light quantity (Hashida et al., 2022), this is not the case universally (Figure 3a). We obtained the full spectrum for each channel, providing insight into optical crosstalk between the waveband channels. In addition to calibration, we encourage plant scientists to more comprehensively report their lighting conditions in controlled environment experiments (Both et al., 2015).

We have demonstrated that our multidimensional algorithm outperforms other approaches, especially at high complexities. Other approaches assume linearity between intensity and measured light quantity (Hashida et al., 2022) and treat each LED channel as an independent variable. Our approach accounts for potential non-linearity by first subsetting to the calibration point that is closest to the target. By using a multidimensional system of linear equations, we can account for the optical crosstalk between different LED channels. This may not be required if the number of active LED channels is low. However, as research moves towards more complex regimes that emulate natural field conditions, there will be a need for methods to reliably programme them.

Additionally, we present an extra refinement step that uses a local random search. This represents a way to provide additional accuracy for light regimes that require it, and is more efficient than previous measure and adjust approaches (Thomas et al., 2020). However, it is still a rather time-consuming step, and so we recommend it only if the extra accuracy is needed and existing methods are not getting users close enough to the target. Though not demonstrated here, a more thorough grid search, where every possible combination of intensities is tested within a narrow range, can be used to further refine the random search. However, the time required to do a grid search increases exponentially with the number of LED channels involved.

The methods presented here provide a way for the plant sciences community to accurately program complex light regimes on LED-based lights. This will allow researchers to investigate more natural regimes where light quantity and wavelength ratios change gradually through the day. This could provide insights into how variation in light quality and quantity entrains the circadian clock; how plants respond photosynthetically to temporary shade from clouds; and how seasonal changes in light quality affect phenological responses. Our approach may also be beneficial for applied research. For example, it may be used to optimise light regimes for indoor controlled environment agriculture and vertical farms. We look forward to seeing how the community uses our package.

## 5. Conclusions

We present a method for accurately programming complex light regimes using multiple LED channels. By using a multistep, multidimensional algorithm, we can account for non-linearity between intensity settings and measured light output, as well as optical crosstalk between different LED channels, outperforming other approaches. The algorithm-predicted intensities can be further refined using a localised random search. This approach will enable the plant sciences community to explore plants' responses to complex light regimes, such as the conditions experienced in the field.

**Open peer review.** To view the open peer review materials for this article, please visit http://doi.org/10.1017/qpb.2026.10041.

**Supplementary material.** The supplementary material for this article can be found at http://doi.org/10.1017/qpb.2026.10041.

**Data availability statement.** The R package (LightFitR) containing the algorithm and usage documentation is available under the GNU General Public License and can be installed from CRAN, using 'install.packages("LightFitR")'. The package code can be viewed on GitHub: https://github.com/ginavong/LightFitR.

The processed data and analysis code (in R) for this paper is available under the GNU General Public License in the 2024_LightFitR_MethodsPaper GitHub repository: https://github.com/ginavong/2024_LightFitR_MethodsPaper.

The raw datasets supporting the conclusions of this article are available on Zenodo: https://doi.org/10.5281/zenodo.15584172.

## Acknowledgements

The authors would like to thank the Horticulture Team at the University of York Biology Department for their maintenance of the growth facilities, cabinets and helpful insights. The authors are also grateful to David Nelmes for IT support with the lights, and Ethan Redmond for stimulating discussion about the mathematics used in this paper.

**Author contributions.** GV and DE designed the algorithm and wrote the R package. PS and JD installed Heliospectra lights into the growth cabinets and made additional modifications to the cabinets. GV, WC, PS and JD measured and tested the lights. GV carried out the experiment design, analysis and figure creation. KD and DE supervised the work. All authors contributed to the writing of this manuscript.

**Funding statement.** The authors would like to thank the Royal Society (RGS\R2\212345: D.E.), Biotechnology and Biological Sciences Research Council (Responsive Mode) (BB/V006665/1: DE) and the Biotechnology and Biological Sciences Research Council White Rose Doctoral Training Partnership (BB/T007222/1: GV).

**Competing interest.** The authors declare none.

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
