## [Reviewer Report]

I. Overall Evaluation

This manuscript introduces a multidimensional algorithm for accurately programming multi-channel LED light regimes and its accompanying R package LightFitR, addressing the critical technical bottleneck of “precisely generating complex dynamic light environments” in controlled-environment plant growth research. The core innovation lies in simultaneously accounting for the LED intensity-irradiance nonlinear relationship and optical crosstalk between spectral channels, significantly outperforming traditional single-channel independent assumption methods. The manuscript holds important theoretical and applied value but exhibits deficiencies such as insufficient hardware universality verification and lack of plant physiological empirical evidence, requiring further revisions and improvements.

II. Major Strengths

1.Theoretical and Methodological Innovation

Proposes a multidimensional calibration algorithm integrating “crosstalk matrix modeling + local optimal solution strategy,” systematically resolving optical crosstalk (spectral overlap) and intensity-irradiance nonlinear response issues in multi-channel LEDs for the first time, breaking through the limitations of traditional single-channel assumptions.

2.Expanded Application Scenarios

Enables accurate simulation of natural dynamic light environments, filling the technical gap of “disconnect between experimental control and natural scenarios” in plant dynamic light response research.

3.Tool Practicality

Develops the accompanying R package LightFitR, publicly available on CRAN, facilitating direct application by the plant science community and lowering the programming barrier for complex light regimes.

III. Major Deficiencies and Areas Needing Improvement

1.Severely Insufficient Verification of Hardware and Algorithm Universality

Hardware Coverage Limitation: All experiments were conducted solely using Heliospectra DYNA fixtures (9 channels), failing to cover mainstream plant lighting brands or different channel configurations. Adaptability to low-channel count systems and severe nonlinear responses cannot be verified.

Algorithm Universality Concerns: Although claimed to be “light-agnostic,” the algorithm was not validated on lighting systems with more severe spectral overlap or significantly different nonlinear characteristics (e.g., other LED brands), leaving the boundaries of universality unclear.

2.Lack of Plant Physiological Empirical Data

Only verifies irradiance prediction accuracy of the algorithm, without linking to plant physiological response data (e.g., photomorphogenesis, photosynthetic rate). This prevents direct demonstration of the practical value of this light control method for plant research.

3.Need for Optimization of Document Content and Structure

Abstract and Introduction: Lack clear background introduction on current LED technology limitations in studying complex plant light response mechanisms, failing to sufficiently highlight research necessity.

Methods Section: Key steps (e.g., optical crosstalk calculation methods, random search implementation details) are insufficiently detailed, affecting experimental reproducibility; algorithm applicability under different light conditions is not comprehensively analyzed or discussed.

Discussion Section: Insufficient depth, failing to thoroughly explore the algorithm’s universality across fixtures, potential impacts on plant science, future research directions, and limitations. Discussion of universality boundaries is inadequate.

IV. Specific Revision Suggestions

1.Strengthen Verification of Hardware and Algorithm Universality

Supplement the discussion or data on hardware and algorithm universality.

Define universality boundaries: Clearly delineate the algorithm’s applicable scope (e.g., channel count, wavelength range, spectral overlap degree) in the Discussion.

2.Supplement Plant Physiological Correlations or Research Value Explanations

Strengthen theoretical basis: Cite plant photomorphogenesis theories (e.g., Low Irradiance Response Pathway) to explain the importance of nonlinear modeling for specific wavelength combination studies, clarifying the algorithm’s irreplaceability in plant physiological research.

3.Optimize Document Content and Structure

Abstract and Introduction: Clearly articulate current technical limitations of LED technology in complex plant light response mechanisms and the specific solutions provided by this research, emphasizing research necessity and conclusions.

Methods Section: Elaborate on key steps (e.g., spectral data collection methods, crosstalk coefficient calculation formulas, and random search implementation details); add analysis of algorithm applicability under different light conditions.

Discussion Section: Supplement potential impacts of the algorithm on plant light signal transduction, photosynthetic physiology, etc. (e.g., advancing dynamic photoperiod research); clarify limitations (e.g., hardware adaptation difficulties) and future optimization directions.

LightFitR Documentation: Provide hardware adaptation guidelines, including custom calibration data formats, crosstalk matrix input templates, and parameter adjustment recommendations for low-channel count fixtures.

V. Concluding Remarks

This research demonstrates significant innovation, with the algorithm addressing key technical bottlenecks in multi-channel LED light control and holding important application value. Acceptance is recommended in principle, but substantial revisions are required as outlined above to enhance universality, rigor, and document clarity. Post-revision re-evaluation will focus on hardware verification, physiological correlations, and document optimization.

---

## [Reviewer Report]

Manuscript ID QPB-2025-0025 entitled “Accurately Programming Complex Light Regimes with Multi-channel LEDs” for publication in Quantitative Plant Biology

This manuscript reports the emission controllability of a commercial LED light source when using a control program that the authors developed for this study. Details of the program algorithm and the emission control performance were described. The program has been released to the plant science community. Thereby, plant light responses are expected to be investigated more carefully and precisely using this or similar LED products. To improve the validity and to enhance the impact of this manuscript, this reviewer suggests that the authors present some aspects of practical dynamic performance and limitations of the proposed LED control method. Specific comments are presented below for use by the authors when revising the manuscript.

1) The phrase “novel agricultural technologies” on line 45 of page 3 is vague. The exact meaning should be presented and described more specifically according to the context of the Introduction.

2) The text on page 3 describes that “these 3 components allow the plant to infer information about the time of day, seasons and shade avoidance.” The phrase “shade avoidance” might be replaced with “shade” in this context to make it more comprehensible and grammatically parallel for readers.

3) On line 154 of page 8, the use of the unit micro-W cm-2 nm-1 s-1 should be reconsidered. It might be replaced with micro-W cm-2 nm-1.

4) The term “intensity” plays a central role in this manuscript, but it has been defined ambiguously without designating any physical unit. The intensity term has been used in the context similarly to “volume” for audio equipment. Can the authors use any other specific term such as “dimming scale” to represent the degree of the light output? In addition, can users change the intensity values (0, 1, 5, 10, 20, 50, 100, 200, 300, 400, 500, 600, 700, 800, 900, and 1000) discretely (digital) or continuously (analog)?

5) To adjust lamp irradiance, electrical control circuits regulate the power input to LED chips. Current amplitude regulation or pulse width modulation (PWM) is often used for LED dimming operations. The latter is common in consumer products. How does the present lamp control system regulate light emissions? If the lamp is operated with the PWM dimming control, then, in principle, the emitted light shapes pulses in the time domain (frequently repeats on / off), which differs from gradual transitions of natural sunlight. This operation might limit the applicability of the lamp system, although the authors imply in some parts of the manuscript, including the Abstract, Discussion, and Conclusion, that their method has potential to approximate natural sunlight conditions.

6) Please define the unit of “irradiance” on page 6. The unit “mol m-2 s-1” used in the manuscript as “irradiance” by the authors is often stated alternatively as “photon flux density” (Zavafer et al., 2023 Biophysical Reviews 15:385–400). Similarly, “spectral photon flux” (Verhoeven, 1996, Glossary of terms used in photochemistry. Pure and Applied Chemistry 68:2275) is often used for the unit “mol m-2 s-1 nm-1” which is also designated as “irradiance” in the present manuscript. Please reconsider the use of the “irradiance” term for representing the physical units of “mol m-2 s-1” and “mol m-2 s-1 nm-1”.

7) The LED lamp configuration (luminaire dimensions, lamp chip numbers and arrangements of each LED type on the luminaire surface, etc.) should be illustrated for readers and explained.

8) The lamp used for this study includes LEDs of nine types, which are characterized by their peak emission wavelength (380, 400, 420, 450, 530, 620, 660, and 735 nm) and color temperature (5700 K). As the authors implied in the Discussion section, natural daylight spectra can be reproduced approximately using combinations of the nine LED type. The manuscript should present for readers some spectra of combined light emitted simultaneously from those LEDs of the nine types.

9) Some dynamic spectral controllability of the LED system should be demonstrated if the authors point out limitations of conventional on/off light controls and advocate the importance of dynamic light controllability for reproducing natural sunlight fluctuations.

---

## [Editor Report]

Dear Authors

On behalf of the Editorial Board, I would like to thank you for submitting your manuscript, “Accurately Programming Complex Light Regimes with Multi-channel LEDs”. Please accept our apologies for the time it has taken to reach a decision regarding your submission; we sincerely thank you for your patience during the review process.

We have now received feedback from two expert reviewers who assessed your work. I am pleased to inform you that their overall assessment is positive. The reviewers have recommended that your manuscript be accepted for publication after minor revisions.

The reviewers' comments are included below. We ask that you carefully address all points raised in your revised manuscript and in a point-by-point response to the reviewers.

Please submit your revised manuscript within four weeks of the date of this letter. If you require additional time, please let us know.

We look forward to receiving your revised manuscript and hope to move forward with its publication.

Sincerely,

Boon Leong Lim

Associate Editor

---

## [Reviewer Report]

Manuscript ID QPB-2025-0025.R1 entitled “Accurately Programming Complex Light Regimes with Multi-channel LEDs” for publication in Quantitative Plant Biology

The authors addressed most of the comments raised by this reviewer. However, some minor concerns remain.

1) The graphical abstract can be replaced with Fig. 2b because “irradiance” has been replaced with “photon flux density” in the revised manuscript.

2) On line 179 of page 10, the use of the unit micro-mol m-2 nm-1 should be reconsidered. It might be replaced with micro-mol m-2 s-1 nm-1.

3) Figure 1 has three panels, (a), (b), and (c), but the caption for Figure 1 only describes two of them.

---

## [Reviewer Report]

The authors have responded to all the questions I raised and made corresponding revisions, which meet the review requirements. I recommend accepting the manuscript.

---

## [Editor Report]

Dear Authors

I am pleased to inform you that your manuscript will be accepted after the following minor revision, as suggested by one of the reviewers:

1) The graphical abstract can be replaced with Fig. 2b because “irradiance” has been replaced with “photon flux density” in the revised manuscript.

2) On line 179 of page 10, the use of the unit micro-mol m-2 nm-1 should be reconsidered. It might be replaced with micro-mol m-2 s-1 nm-1.

3) Figure 1 has three panels, (a), (b), and (c), but the caption for Figure 1 only describes two of them. 

Yours sincerely

Editor